# Pathogenesis Underlying Neurological Manifestations of Long COVID Syndrome and Potential Therapeutics

**DOI:** 10.3390/cells12050816

**Published:** 2023-03-06

**Authors:** Albert Leng, Manuj Shah, Syed Ameen Ahmad, Lavienraj Premraj, Karin Wildi, Gianluigi Li Bassi, Carlos A. Pardo, Alex Choi, Sung-Min Cho

**Affiliations:** 1Department of Surgery, Johns Hopkins University School of Medicine, Baltimore, MD 21205, USA; 2Department of Neurology, Johns Hopkins University School of Medicine, Baltimore, MD 21205, USA; 3Department of Neurology, Griffith University School of Medicine, Gold Coast, Brisbane, QLD 4215, Australia; 4Critical Care Research Group, The Prince Charles Hospital, Brisbane, QLD 4032, Australia; 5Faculty of Medicine, University of Queensland, Brisbane, QLD 4072, Australia; 6Institute of Health and Biomedical Innovation, Queensland University of Technology, Brisbane, QLD 4000, Australia; 7Intensive Care Unit, St Andrew’s War Memorial Hospital and the Wesley Hospital, Uniting Care Hospitals, Brisbane, QLD 4000, Australia; 8Wesley Medical Research, Auchenflower, QLD 4066, Australia; 9Department of Pathology, Johns Hopkins University School of Medicine, Baltimore, MD 21205, USA; 10Division of Neurosciences Critical Care, Department of Neurosurgery, UT Houston, Houston, TX 77030, USA; 11Divisions of Neurosciences Critical Care and Cardiac Surgery, Departments of Neurology, Surgery, Anesthesiology and Critical Care Medicine and Neurosurgery, Johns Hopkins University School of Medicine, Baltimore, MD 21205, USA

**Keywords:** COVID-19, SARS-CoV-2, long COVID, neurological manifestations, neurological complication, outcome, brain fog

## Abstract

The development of long-term symptoms of coronavirus disease 2019 (COVID-19) more than four weeks after primary infection, termed “long COVID” or post-acute sequela of COVID-19 (PASC), can implicate persistent neurological complications in up to one third of patients and present as fatigue, “brain fog”, headaches, cognitive impairment, dysautonomia, neuropsychiatric symptoms, anosmia, hypogeusia, and peripheral neuropathy. Pathogenic mechanisms of these symptoms of long COVID remain largely unclear; however, several hypotheses implicate both nervous system and systemic pathogenic mechanisms such as SARS-CoV2 viral persistence and neuroinvasion, abnormal immunological response, autoimmunity, coagulopathies, and endotheliopathy. Outside of the CNS, SARS-CoV-2 can invade the support and stem cells of the olfactory epithelium leading to persistent alterations to olfactory function. SARS-CoV-2 infection may induce abnormalities in innate and adaptive immunity including monocyte expansion, T-cell exhaustion, and prolonged cytokine release, which may cause neuroinflammatory responses and microglia activation, white matter abnormalities, and microvascular changes. Additionally, microvascular clot formation can occlude capillaries and endotheliopathy, due to SARS-CoV-2 protease activity and complement activation, can contribute to hypoxic neuronal injury and blood–brain barrier dysfunction, respectively. Current therapeutics target pathological mechanisms by employing antivirals, decreasing inflammation, and promoting olfactory epithelium regeneration. Thus, from laboratory evidence and clinical trials in the literature, we sought to synthesize the pathophysiological pathways underlying neurological symptoms of long COVID and potential therapeutics.

## 1. Introduction

Coronavirus disease 2019 (COVID-19) is a multi-system disease caused by infection with severe acute respiratory syndrome coronavirus 2 (SARS-CoV-2). The Centers for Disease Control and Prevention (CDC) considers long COVID-19 to be present when symptoms last longer than four weeks after the initial infection. The collection of symptoms goes by many names, including but not limited to “long COVID,” “chronic COVID,” “post-acute sequelae of COVID-19,” and “post-COVID conditions.” Prior studies have reported different frequencies of long COVID, ranging from 13.3% to 54% of patients after initial SARS-CoV-2 infection [1,2]. Notably, one subtype of long COVID includes neurological sequelae, which some reports have identified as being present in one third of patients in the first six months following acute COVID-19 infection. These symptoms manifest as objective abnormalities on neurological examination, such as motor/sensory deficits, hyposmia, cognitive deficits, and postural tremor [3].

While prior studies have proposed potential mechanisms for the symptoms of long COVID, there are limited reports that synthesize and evaluate the pathophysiology of neurological manifestations of long COVID and its therapeutic options. In doing so, we connect basic science research, translational research, and the findings of epidemiological and clinical studies.

## 2. Clinical Evidence of Neurological Involvement in Long COVID

### 2.1. Epidemiology

In a meta-analysis of 257,348 COVID-19 patients, some of the most common long COVID symptoms at three to six months included fatigue (32%), dyspnea (25%), and concentration difficulty (22%), reflecting the multisystemic nature of long COVID [4].

In addition to these symptoms, there is a specific cluster of specific neurological symptoms and sequelae of long COVID. For instance, in a sample of 10,530 long COVID patients at a 12-week follow-up, some of the most common neurological symptoms included fatigue (37%), brain fog (32%), memory issues (28%), attention disorder (22%), myalgia (28%), anosmia (12%), dysgeusia (10%), and headaches (15%) [5]. Some of these symptoms continue to persist at longer follow-up periods—including six-month and one-year follow-ups after initial diagnosis [6,7,8]. Considering these studies, it is evident that cognitive symptoms, headaches, sleep disorders, neuropathies, and autonomic dysfunction are some of the most common neurological manifestations of long COVID. Other, less frequent, neurological sequelae include dysexecutive syndrome, ataxia, and motor disturbances [9,10]. In all, these symptoms can lead to significant dysfunction and disability, with around 30% of long COVID patients aged 30–59 indicating that their neurological symptoms made them severely unable to function at work [10].

### 2.2. Risk Factors

Specific demographic risk factors for long COVID have been identified. Females were reported to have a higher risk of developing long COVID symptomatology [1,11,12,13,14,15]. Information on age is less unanimous. Several studies have reported that older patients (vs. younger) are at increased risk of developing long COVID [1,14,15,16,17]. However, other studies have shown that younger patients are at increased risk, while some studies have shown no association between age and the development of long COVID [11,12,16,18]. Regarding race/ethnicity, a study of 8325 patients with long COVID reported that non-Hispanic white patients were more likely to develop long COVID while non-Hispanic black patients were less likely to develop long COVID [14]. Alternatively, in a longitudinal analysis of 1038 patients, race/ethnicity had no significant association with long COVID occurrence [17]. 

However, there are sparse data on the specific risk factors for neurological manifestations of long COVID. In one study, female sex and older age were shown to be associated with the neurological manifestations of long COVID, while race/ethnicity, COVID-19 severity, and other comorbidities, such as hypertension, diabetes, and congestive heart failure, were not [19]. Additionally, it has been noted that an increased severity of neurological symptoms is associated with a diminished CD4^+^ T cell response against the spike protein, suggesting that the T cell response is necessary to counteract the severity of neurological long COVID. In this cohort, mRNA COVID-19 vaccination elevated the T cell response and helped diminish the severity of neurological symptoms in long COVID [20]. 

Yet overall, the impact of demographic factors such as race/ethnicity, as well as comorbidities on long COVID, needs more dedicated epidemiological studies as many of the previous studies are influenced by geographical and recruitment biases.

Focusing on biological and medical factors influencing long COVID, studies have focused on the magnitude and severity derived from the acute phase of COVID-19. For patients who required intensive care unit (ICU) admission in the acute phase of COVID-19, long-term impairment following ICU discharge appears to be frequent. For example, in a study of 117 patients that required high-flow nasal cannulae, non-invasive mechanical ventilation, or invasive mechanical ventilation, 86% reported long COVID symptoms at a six-month follow-up. These included, but were not limited to, fatigue, muscle weakness, sleep difficulties, and smell/taste disorders [6,17]. Metabolic risk factors such as a high body mass index, the presence of insulin resistance, and diabetes mellitus have been associated with long COVID [14,15,16,21,22]. Not surprisingly, patients who were “hospitalized” in the acute phase of COVID-19 were more likely to develop long COVID symptoms [15,17]. Other risk factors include Epstein–Barr virus reactivation, history of smoking, exposure to air toxicants and pollutants, and the presence of chronic comorbid conditions [13,14,20,22,23]. 

### 2.3. Outcomes

An increased risk of mortality has been observed in COVID-19 patients with post-acute sequelae (defined as at least 365 days of follow-up time for long COVID symptoms). In a large study of 13,638 patients, an increased 12-month mortality risk after recovery from the initial infection was observed as compared with patients with suspected COVID-19 and who had a negative polymerase chain reaction (PCR) test [24]. Additionally, long COVID patients with more severe initial infections (defined by occurrence of hospitalization) had an increased 12-month mortality risk after recovery from the initial infection and subsequent development of post-acute sequelae in comparison with patients with moderate or mild initial COVID-19 infections [24]. Furthermore, age, male sex, unvaccinated status, and baseline comorbidities were associated with higher mortality in patients with long COVID when followed over time [25]. Regarding vaccination, a systematic review of 989,174 patients across different studies demonstrated that vaccination before acute COVID-19 infection was associated with a reduced risk (RR = 0.71) of developing non-neurological symptoms of long COVID [26]. Likewise, in a survey of long COVID patients who had not yet been vaccinated, most patients had an improved average symptom score, suggesting that vaccination may play a role in mitigating the symptoms of long COVID [27]. 

Much of the available literature focuses on mortality outcomes related to long COVID broadly. To our knowledge, there are no studies that report mortality outcomes on the neurological symptoms of long COVID specifically. 

## 3. Mechanisms of Neurological Long COVID and Review of Therapeutics

### 3.1. Viral Neuroinvasion and Persistent Viral Shedding

The SARS-CoV-2 virus is known to invade human cells through engagement with specific membrane cell receptors which include angiotensin-converting enzyme 2 (ACE2) transmembrane receptor and activation of SARS-CoV-2 spike protein by transmembrane serine protease 2 (TMPRSS2) cleavage. Undoubtedly, polymorphisms that alter the ACE2 and spike protein interaction, the TMPRSS2 proteolytic cleavage site, and ACE2 expression correlate with the susceptibility and severity of COVID-19 with some ACE2 variants incurring up to a three-fold increase in the development of severe disease [28,29]. Since the severity of disease is associated with the incidence of long COVID symptoms [30], there is a possibility that ACE2 and TMPRSS2 polymorphisms could potentiate long COVID as well. To our knowledge, the only study to have investigated this relationship found no predisposition of formerly identified ACE2 and TMPRSS2 polymorphisms linked to disease severity for long COVID symptoms in patients who were previously hospitalized for COVID-19 [31].

Relating specifically to neurological symptoms of long COVID, receptors are expressed by endothelial and nervous system cells such as neurons, astrocytes, and oligodendrocytes [32,33,34,35,36]. However, the possibility for viral invasion of neural tissue remains highly debated. The presence of SARS-CoV-2 in cortical neurons from autopsy studies and replicative potential of SARS-CoV-2 in human brain organoids implicates the neurotropic effects of the virus in the pathogenesis of neurological symptoms to some extent [36], but the possibility and mechanism of direct viral infection of the central nervous system (CNS) still remain unclear. Current proposed pathways include transsynaptic invasion by transport along the olfactory tract [37], which is highly unlikely due to the lack of ACE2 receptors and TMPRSS2 on olfactory neurons [38,39,40], and hematogenous spread through invasion of choroid plexus cells and pericytes [41,42]. The latter has been shown to occur in human neural organoid models where ACE2 receptors are heavily expressed on the apical side of the choroid epithelium, allowing for SARS-CoV-2 invasion through the vasculature, subsequent ependymal cell death, and blood–CSF barrier (B-CSF-B) disruption [41]. Despite this potential for viral neuroinvasion through hematogenous means, there is overwhelming evidence showing a lack of SARS-CoV-2 RNA and protein in the cerebrospinal fluid (CSF) of COVID-19 patients with neurological symptoms [43,44], globally in the brain tissue from autopsy studies [45,46], and even within the choroid plexus of individuals with severe disease [47].

#### 3.1.1. Invasion of Olfactory Epithelium

In contrast, persistent anosmia is a symptom of long COVID that likely results from lasting effects of direct viral damage of the olfactory epithelium. In acute COVID-19, SARS-CoV-2 can infect non-neural cell types that express ACE2 in the olfactory epithelium, specifically stem cells, perivascular cells, sustentacular cells, and Bowman’s gland cells (Figure 1), that leads to cell death and loss of uniformity demonstrated in humanized ACE2 mice [38,48]. Unlike with transient anosmia observed in other respiratory infections, imaging studies performed on patients with persistent COVID-19 anosmia demonstrated extensive damage to the olfactory epithelium; this manifests as thinning of olfactory filia and reduction in olfactory bulb volumes [39,49]. Additionally, biopsies of olfactory mucosa from patients with persistent anosmia of varying etiologies reinforce the connection between thinning of olfactory epithelium and enduring symptoms [50]. Thus, the loss of stem cells and support cells in the neuroepithelium causes failure of epithelial repair, resulting in the thinning and loss of olfactory dendrites likely accounting for long-standing anosmia [51]. In addition, persistent inflammation evidenced by elevated levels of interleukin 6 (IL-6), type I interferon (IFN), and C-X-C motif chemokine ligand 10 (CXCL10) within the olfactory epithelium, secondary to invasion, appears to contribute to long-standing anosmia [52]. Altogether, SARS-CoV-2 invasion of supportive cells and subsequent long-lasting local inflammation causes irreversible damage to the olfactory epithelium and are thus the main drivers of persistent hyposmia, anosmia, and dysgeusia.

The invasion of the olfactory epithelium by SARS-CoV-2, however, does not necessarily serve as a window of opportunity for neuroinvasion. Although early in vitro and in vivo studies may suggest the possibility for neuroinvasion of the CNS in the pathogenesis of disease in long COVID, they are limited by neuropathological evidence for SARS-CoV-2 in the brain parenchyma or CSF of patients. Additionally, anatomical barriers to neuroinvasion, such as the perineurial olfactory nerve fibroblasts that wrap olfactory axon fascicles and the added absence of ACE2 receptors for entry on olfactory neurons [51], further call into question the feasibility for this mechanism of pathogenesis. Thus, to this date, there has not been a validated demonstration of SARS-CoV-2 invasion of, and replication in, the CNS. 

#### 3.1.2. Dysbiosis and Brain–Gut Axis

Following SARS-CoV-2 infection, viral shedding has been shown to persist in the upper respiratory and gastrointestinal (GI) epithelium for a median of 30.9 days and 32.5 days, respectively, in severe COVID-19 [53,54]. Due to the ability of SARS-CoV-2 to cause persistent symbiont depletion and gut dysbiosis in the GI tract [55], prolonged viral shedding maintains microbiome perturbations, likely resulting in brain–gut axis dysfunction [56]. Additional evidence of brain–gut axis alteration from a co-expression analysis in SARS-CoV-2-infected human intestinal organoids revealed ACE2 co-regulation of dopa-decarboxylase (DDC) and clusters of genes involved in the dopamine metabolic pathway and absorption of amino acid precursors to neurotransmitters (Figure 1) [57]. Thus, the involvement of gut dysbiosis and brain–gut axis alterations that persist due to continual viral shedding have been suggested as a possible mechanism in neurological manifestations of long COVID [7].

#### 3.1.3. Reactivation of Herpesviruses

Aside from SARS-CoV-2 viral persistence, reactivation of viruses of the herpesviridae family, including Epstein–Barr virus (EBV) and Varicella-zoster virus (VZV), have also been well documented in long COVID patients. EBV and VZV, a lymphotropic gammaherpesvirus and neurotrophic alphaherpesvirus respectively, can independently affect more than 90% of people worldwide [58,59]. Both viruses can remain latent in host cells after primary infection (in memory B cells in EBV and the neurons of sensory ganglia in VZV) such that the onset of a stressor, such as another acute viral infection, can lead to the reactivation of these herpes viruses and cause inflammation and neurological symptoms. SARS-CoV-2 can act as that stressor and precipitate reactivation of other viruses in COVID-19 and long COVID symptomatology. 

According to an early retrospective study of acute COVID patients post-hospitalization, 25% of patients with severe disease had increased serological titers of early antigen IgG (EA-IgG) and viral capsid antigen IgG (VCA-IgG) which serve as proxy markers for reactivation of EBV [60]. More specific to long COVID, a survey study found that two thirds of patients with symptoms 90 days after primary SARS-CoV-2 infection were positive for EBV reactivation, which was also indicated by positive titers of VCA-IgG and early antigen-diffuse IgG (EA-D IgG) [61]. Higher frequency of long COVID symptoms experienced by patients were also significantly correlated with increased EA-IgG titers. Similarly, a longitudinal study of 309 patients tracked from primary infection to convalescence revealed EBV viremia to be one of the four main risk factors for developing long COVID symptoms, with the other three being type II diabetes, SARS-CoV-2 RNAemia, and autoantibodies formation [20]. EBV reactivation has been specifically associated with memory and fatigue in long COVID. Apart from COVID-19, the immune response to EBV reactivation has been shown to reflect that of myalgic encephalomyelitis (ME) or chronic fatigue syndrome (CFS) which could link EBV viremia to the development of ME/CFS-like symptoms in long COVID [62,63]. This immune profile has been identified in a cross-sectional study with 215 long COVID patients where there was an elevated antibody reactivity to EBV gp23, gp42, and EA-D which all were correlated with interleukin 4 (IL-4) and IL-6 producing CD4^+^ T cells [64]. 

The same study also identified significant levels of antibody reactivity to the VZV glycoprotein E which was similarly associated with the immune profile mentioned above. VZV manifestations are also common in COVID-19, occurring in about 17.9% of patients mostly and in the form of dermatome rashes, with rare instances of encephalitis-meningitis and vasculitis [65]. Although less prominent in long COVID pathogenesis than EBV reactivation, VZV reactivation can still contribute to neurological symptoms due to its involvement with the CNS. 

#### 3.1.4. Related Long COVID Therapeutics: Antivirals

The persistence of SARS-CoV-2 through viral shedding provides an impetus to the consideration of antiviral drugs as potential long COVID therapies. Antiviral drugs used in the treatment of acute COVID-19, specifically remdesivir, molnupiravir, fluvoxamine and nirmatrelvir/ritonavir combination (Paxlovid), have demonstrated substantial reductions in mortality and hospitalization [66,67]. Unlike its counterparts, nirmatrelvir is a highly specific competitive inhibitor of the SARS-CoV-2-3CL protease and therefore of its viral replication. Ritonavir increases the bioavailability of nirmatrelvir by preventing its hepatic metabolism [68]. An RCT in high-risk non-hospitalized adults with acute COVID-19 showed that Paxlovid reduced viral load rapidly (by day five) as well as mortality and hospitalization (Table 1) [67]. Paxlovid may be vital in reducing neurological symptoms of long COVID, secondary to viral load/shedding, such as persistent cognitive impairment and insomnia. In a recent study (preprint), Xie et al. compared patients who received no antiviral or antibody treatment during acute COVID-19 infection (N = 47,123) with those treated with oral nirmatrelvir within five days of a positive COVID-19 test (N = 9217). Compared with the controls, patients acutely treated with nirmatrelvir had substantially lower risk of developing long COVID symptoms [69]. Their definition of long COVID symptomatology involved 12 outcomes, including myalgia and neurocognitive impairment [69]. While the exact indications for its use in long COVID patients remain to be defined by RCT protocols (Table 2, NCT05576662), Paxlovid demonstrates promise. We posit that Paxlovid may be useful in reducing persistent viral shedding from infected epithelium, and can therefore reduce mechanisms secondary to SARS-CoV-2 invasion such as the previously mentioned gut dysbiosis and inflammation of the olfactory mucosa. 

The use of antivirals to address the issue of viral reactivation in long COVID, including treatments for herpesviruses such as acyclovir and valacyclovir, has been documented but their efficacy in alleviating neurological symptoms of long COVID have yet to be assessed [65]. One retrospective study in Wuhan assessed 28-day mortality outcomes for 88 COVID-19 patients with EBV reactivation that were treated with ganciclovir compared with those of matched controls [60]. Ganciclovir treated patients had a significantly higher survival rate than controls, however, specific neurological symptoms were not assessed. Further studies are needed to demonstrate the effectiveness of antivirals specific to herpesvirus reactivation in long COVID patients.

In addition to the specific use of antivirals, other agents, such as cannabidiol, may have some antiviral efficacy as a therapy for long COVID. It is known that the active metabolite in cannabidiol, 7-OH-CBD, can block SARS-CoV-2 replication by inhibiting viral gene expression, upregulating interferon expression, and promoting antiviral signaling pathways [76]. Notably, cannabidiol has been reported to downregulate ACE2 and TMPRSS2 [77]—key enzymes involved in the SARS-CoV-2 virus invasion process and the potential evolution to long COVID. A phase 2 clinical trial (NCT04997395) has begun looking into the feasibility of using cannabidiol as a treatment for long COVID (Table 2). Additionally, Cannabidiol has been shown to induce neuroprotective effects [78,79]. Taken together, this suggests that cannabidiol can help ameliorate the neurological symptoms of long COVID, although future clinical trials are needed to provide further evidence.

#### 3.1.5. Related Long COVID Therapies: Anosmia

Regeneration of olfactory mucosa occurred with administration of intranasal insulin in non-COVID-19 patients. Insulin, through its action as a phosphodiesterase enzyme inhibitor, can increase cyclic adenylate monophosphate (cAMP) and cyclic guanylate monophosphate (cGMP) levels by interacting with the nitric oxide cycle [80]. These growth factors are known to stimulate the olfactory epithelium and promote regeneration [80]. In their pragmatic randomized controlled trial (RCT) in a small population (N = 38), Rezaian et al. assessed the success of twice weekly intranasal protamine insulin gel foam therapy (vs. normal saline placebo) on patients with mild to severe post-infectious hyposmia (Table 1) [74]. They determined that olfaction (via Connecticut Chemosensory Clinical Research Centre score) at four weeks was notably better in the insulin treated groups (5.0 ± 0.7) versus the placebo group (3.8 ± 1.1, *p* < 0.05) [74]. A larger trial in COVID-19 patients is ongoing (Table 2, NCT05104424). More novel interventions have also demonstrated promise. Indeed, 40 patients with anosmia post COVID-19 infection were randomized to either intranasal insulin film vs. normal saline (placebo). At 30 min after administration, patients in the treatment group had significantly greater odor detection compared with both their baseline and the placebo group [73]. Although RCTs with larger sample sizes and longer follow-up are necessary, these findings are promising in treating persistent anosmia in long COVID. 

Due to the damaging effects of local inflammation on the olfactory epithelium, corticosteroids have been used in certain patients to hasten the recovery and repopulation of the olfactory epithelium [71,81]. In 2021, an RCT of mometasone nasal spray, which included one hundred COVID-19 patients with post-infection anosmia, assigned patients to two treatment branches: mometasone furoate nasal spray with olfactory training for three weeks (N = 50) or the control group with only olfactory training (N = 50). There was no significant difference in duration of smell loss, from anosmia onset to self-reported complete recovery, between groups (*p* = 0.31). However, a significant improvement in smell score was recorded in both groups by week three [71]. Despite this, Singh et al. were able to demonstrate significant improvements in smell (on day five) compared with baseline (day one) using fluticasone nasal spray compared with no intervention (Table 1) [70]. 

It should be noted that many agents for the treatment of postinfectious hyposmia have been studied previously for non-COVID-19 patients including pentoxifylline, caffeine, theophylline, statins, minocycline, zinc, intranasal vitamin A, omega-3, and melatonin. An in-depth evaluation of their use in non-COVID-19 anosmia is outside the scope of this review. However, in their detailed systematic review, Khani et al. posit that different combinations of the above agents may be of use in long COVID depending on the etiology (viral invasion vs. inflammatory damage) [81].

### 3.2. Abnormal Systemic and Neurologic Immunological Response

With a unique range of immune cell phenotypes, chemokine and cytokine production, and inflammatory molecules, the immunological response to SARS-CoV-2 infection has been widely investigated to rationalize some of the neurological symptoms of long COVID. Although autoantibody generation has been proposed, this inflammatory response has been better characterized with persistent systemic inflammation leading to expansion of monocyte subsets and T cell dysregulation, which in turn is associated with BBB dysfunction, neural glial cell reactivity, and subcortical white matter demyelination (Figure 2).

#### 3.2.1. Systemic Inflammation

After SARS-CoV-2 infection, a variety of systemic inflammatory processes are upregulated, and specific immune cell populations are expanded; this disturbance of the peripheral immune system can persist for many months after the infection, which can lead to neurological symptoms. Broadly speaking, when compared with healthy controls, recovered COVID-19 individuals exhibited differences in the populations of innate immune cells, such as natural killer cells, mast cells, and C-X-C motif chemokine receptor 3^+^ (CXCR3^+^) macrophages, as well as adaptive immune cells, such as T-helper cells and regulatory T cells ([82], p. 2). With non-naive phenotypes, these cells tend to secrete and be activated by increased levels of cytokines and inflammatory markers, including, but not limited, to interferon β (IFN-β), interferon λ1 (IFN-λ1), C-X-C motif chemokine ligand 8 (CXCL8), C-X-C motif chemokine ligand 9 (CXCL9), CXCL10, interleukin 2 (IL-2), IL-6, and interleukin 17 (IL-17) [83,84]. Several studies have drawn a striking similarity between the symptomatology of long COVID and mast cell activation syndrome (MCAS), wherein aberrant mast cell activation promotes excessive release of inflammatory mediators, such as type 1 IFNs, and cytokine activation of microglia [85,86]. Triggered by viral entry, these mast cells are commonly found at tissue–environment interfaces and may contribute to persistent systemic inflammation microvascular dysfunction with CNS disturbances in long COVID [85,86]. SARS-CoV-2 specific T cell responses have also been described to have increased breadth and magnitude. These signature T cell responses against SARS-CoV-2 increase with higher viral loads, indicated by significantly elevated levels of nucleocapsid-specific interferon gamma (IFN-γ) producing CD8^+^ cells in serum samples of patients with persistent SARS-CoV-2 PCR positivity [87,88].

Furthermore, a study using flow cytometry on the peripheral blood of long COVID patients detected elevated expansion of non-classical monocytes (CD14^dim^CD16^+^) and intermediate monocytes (CD14^+^CD16^+^) up to 15 months post infection when compared with healthy controls [89]. Physiologically, non-classical monocytes are involved in complement-mediated and antibody-dependent cellular phagocytosis against viral insults and are commonly found along the luminal side of vascular endothelium, thereby contributing to BBB integrity. For example, severe long COVID patients were found to have increased levels of macrophage scavenger receptor 1 (MSR1), signifying a high degree of peripheral macrophage activation that can in turn disrupt the BBB and cause tissue damage [90]. Alternatively, intermediate monocytes specialize in antigen presentation and simultaneous secretion of pro-inflammatory cytokines. Although this marked systemic hyperinflammatory state has not been shown to directly cause neuropsychiatric manifestations, it may contribute to disease progression via chronic activation of specific monocyte and T cell populations and neurovascular dysfunction of the BBB. These mechanisms can result in the spread of inflammatory molecules and immune cells from the periphery into the CNS, inducing a persistent neuroinflammatory response.

#### 3.2.2. Monocyte Expansion and T Cell Dysfunction

With several parallels to the described systemic inflammation, the CNS exhibits persisting trends of monocyte expansion and T cell exhaustion after SARS-CoV-2 infection, the latter of which refers to T cells adopting a distinct cytokine profile with poor effector function. Similar to systemic trends, monocyte expansion in the CNS refers to an increase in the population of non-classical monocytes (CD14^dim^CD16^+^) and intermediate monocytes (CD14^+^CD16^+^) in CSF of long COVID patients [91,92]. Indeed, monocyte pools analyzed in long COVID patients exhibit a reduction of classical monocytes, indicated by lower levels of pan-monocytic markers and CNS border-associated macrophage phenotypes [91]. While the function of monocytes may be less understood in a chronic disease setting, the expansion of monocyte subsets with antiviral and antigen-presenting phenotypes may be implicated in long COVID symptoms due to its role in BBB disruption and neuroinflammation.

As a result of chronic stimulation by antigens, T cells can assume a distinct cytokine profile with increased inhibitory transcription factor expression and decreased effector function, a process termed exhaustion. More common in CD8^+^ T cells, this signature process implicates phenotypic and functional defects that can limit T cell functional responsiveness in clearing infection in chronic settings [93]. Persisting for months post infection to prevent recurring illness, CD8^+^ memory T cells in serum samples of long COVID patients were found to increase in number with higher levels of cytolytic granule expression but with limited breadth and reduced antigen-specific activation [94]. Although the secretion of cytolytic granzymes, IL-6, and nucleocapsid-specific IFN-γ increase in long COVID, these T cells present with limited polyfunctionality and decreased diversity of effector expression; this altered profile was strongly associated with symptoms of depression and decreased executive function [94]. Regarding localization, this population of T cells producing higher granzyme levels can be seen in certain anatomic niches, such as in microglial nodules which are hotspots for immune response activation [95]. Cytotoxic CD8^+^ T cells also congregate near vasculature and produce a surge of cytokines that disrupts the BBB, causing vascular leakage and the propagation of inflammation [95,96]. Evidence against T cell expansion and exhaustion also exists, as one immunophenotyping study shows that persistent T cell changes and neurological deficits are associated with age rather than ongoing illness and fatigue [92]. 

In summary, though there is a degree of heterogeneity in respect to inflammatory molecules and immune cell populations, long COVID patients with neurological symptoms exhibit persistent systemic inflammation with pronounced differences in circulating myeloid and lymphocyte populations, including prominent peripheral B cell activation with a greater humoral response against SARS-CoV-2 [64]. Elevated levels of non-classical monocytes and intermediate monocytes can bring about altered vascular homeostasis and chronic inflammatory processes, which are largely mediated by Th1 cytokines. Increased amounts of exhausted CD4+ and CD8+ T cells with decreased central memory CD4+ T cells implicate a distinct immunological signature with decreased effector function and resulting aberrant immune engagement [64].

#### 3.2.3. Autoantibody Generation

Autoantibody generation has been hypothesized to contribute to the persisting abnormal immunological response observed post infection. Rather than being caused by the virus directly, the autoimmune antibody reaction is suggested to be a product of the pronounced immune and inflammatory reaction [22,97]. The serologically detected autoantibodies can be categorized into antibodies against extracellular, cell surface and membrane, or intracellular targets, which include immunoglobulin G (IgG) and immunoglobulin A (IgA) antibodies against cytokines [98], angiotensin converting enzyme 2 (anti-ACE2) [99], and nuclear proteins (ANA), respectively [100]. 

Following activation of B cells in the periphery and cytokine abnormalities, these serologic IgG and IgA antibodies exhibit a polyclonal distribution, affect cytokine function and endothelial integrity, and can enter the CNS given the BBB disruption [90]. Although there are limited reports, these autoimmune responses have been proposed to be present with acute-onset encephalitis, seizures, meningitis, polyradiculitis, myelopathy, and neuropsychiatric symptoms [101,102,103,104,105]. Persistent ANA autoreactivity has been linked with long-term symptoms of dyspnea, fatigue, and brain fog seen in long COVID [106]. Anti-ACE2 antibodies, which are associated with fatigue and myelitis, can elicit an abnormal renin–angiotensin response, cause malignant hypertension-related ischemia and upregulate thrombo-inflammatory pathways [97]. While these antibodies have been associated with neurological manifestations after SARS-CoV-2 infection, they have also been limited to parainfection and acute post-infection time periods. Furthermore, small studies have reported the lack of autoantibodies in acute COVID-19 patients presenting with encephalitis [107]. More convincingly, a recent exploratory, cross sectional study illustrated that despite patients exhibiting an array of autoantigen reactivities, the total levels of autoantibodies were definitively not elevated in the extracellular proteome of patients with long COVID compared with convalescent controls [64].

Perhaps sparked by previous demonstrations of peripheral B cell activation, research supporting autoantibody generation in the neurological manifestations of long COVID have been mostly limited to various case reports and studies that, due to sample size and timescale constraints, have limited generalizability. Though autoantibodies can drive inflammation, neuronal dysfunction, and subsequent neurodegeneration, which are all observed in long COVID, this mechanism is not as well understood and warrants further investigations to implicate it in the pathogenesis of long COVID.

#### 3.2.4. Related Long COVID Therapies: Anti-Inflammatory Therapy

Control of inflammation post-infection may attenuate persistent cytokine release, immune cell activation, and the pronounced neural immune response, thereby alleviating neurological symptoms of long COVID. Support for this rationale derives from the studies that showed lower prevalence of long COVID in those with less robust inflammatory and immune responses to acute infection, such as vaccinated patients [69,108] and those treated with antivirals [109,110]. Here, we review several promising anti-inflammatory therapies for those with long COVID. 

A currently recruiting double-blinded placebo-controlled RCT assessing the efficacy of oral lithium (10 mg daily) aims to determine if fatigue, brain fog, anxiety and cognitive outcome scores improve after three weeks of lithium therapy (NCT05618587). Despite the anti-inflammatory effects of lithium, good CNS penetrance, and efficacy in reducing inflammation in patients with acute COVID-19 [111], lithium’s efficacy and benefit–risk profile in patients with long COVID and neurological symptoms have not been proven. 

RSLV-132 is a novel RNase fusion protein that digests ribonucleic acid contained in autoantibodies and immune complexes generated by the humoral immune response. Therefore, RSLV-132 has applications in both autoimmune disease and post-viral syndromes caused by autoantibody generation, such as long COVID. When compared with the placebo in patients with Sjogren’s syndrome, RSLV-132 decreased fatigue, which was assessed using Functional Assessment of Chronic Illness Therapy (FACIT), Fatigue Visual Analogue Score, and Profile of Fatigue Score at week 14 [112]. The phase 2 clinical trial of RSLV-132 (NCT04944121) follows patients 10 weeks after the start of treatment and assesses fatigue using Patient-Reported Outcome Measurement Information System (PROMIS), FACIT scores and long COVID symptoms via questionnaires. The precise indication for RSLV-132 requires further study, as patients with long COVID may have sub-threshold or absent autoantibody levels as previously mentioned [107]. 

RCTs assessing the efficacy of conventional anti-inflammatory agents, such as steroids or IV immunoglobulin, are ongoing: one upcoming trial involves the randomized treatment of patients with either IV methylprednisolone, IV immunoglobulin or IV saline (NCT05350774). Depression, anxiety, and cognitive assessment scores will be compared at the end of the three-month (minimum) follow-up period. Ongoing RCTs of anti-inflammatory agents are summarized in Table 2.

Additionally, brain fog and fatigue, which are the most prevalent neurological symptoms of long COVID [5], might arise from the prolonged neuroinflammation secondary to the innate immune activation (immune cell migration across the BBB and chemokine release) and humoral activation (autoantibody generation) described above. While current trials are investigating various agents that control the systemic inflammation discussed above, dextroamphetamine–amphetamine (NCT05597722) may be suitable for use in long COVID patients to improve brain fog, given its use in attention-deficit/hyperactivity disorder. In a similar vein, low-dose naltrexone has shown promise in improving brain fog and fatigue (self-reported via questionnaire) [72]; an upcoming placebo-controlled RCT (NCT05430152) will provide greater clarity on its efficacy. Along with other mast cell mediator blockers and stabilizers used in MCAS that target mast cell overactivation and subsequent inflammation, antihistamine treatment via a combined histamine H1/H2 receptor blockade has been associated with significant symptomatic improvement in long COVID patients according to a recent observational study [75]. However, further studies are required to determine the optimal patients for the above interventions.

#### 3.2.5. Neural Glial Cell Reactivity

One of the most prominent hypothesized mechanisms of long COVID symptomatology involves the activation of the neuroimmune system through the interplay of neural cells and glial cells, namely astrocytes, microglia, and oligodendrocytes. Astrocytes are critical for CNS homeostasis as they play roles in neuron–glial cell interaction, synaptic function, and blood–brain barrier integrity. Microglia are fundamental for processes of innate immunity within the CNS and are central to synaptic function, maintaining neural networks, and supporting homeostatic repair mechanisms upon injury to the micro-environment. However, with altered cytokine activity and brain injury, glial cells can become overactivated. Evidenced by increased levels of ezrin (EZR) in long COVID patients, these reactive astrocytes upregulate NF-κB, which can cause endothelial cell death and increase extracellular glutamate, resulting in BBB disruption and hyperexcitability-induced neurodegeneration, respectively [90,113,114,115]. Similarly, it is suspected that reactive microglia lose their plasticity-promoting function and facilitate disruption of neural circuitry with the release of microglial cytokines.

Patients with neurological symptoms of long COVID were also found to have increased levels of the C-C motif chemokine ligand 11 (CCL11), an immunoregulatory chemokine that can recruit eosinophils, cross the BBB, induce microglial migration, disrupt hippocampal neurogenesis, and cause cognitive dysfunction (e.g., brain fog) [45,116]. Decreased ramification of microglia is partially stimulated by CCL11, causing the release of microglial cytokines and the death of vulnerable neuroglial cells, such as the myelinating oligodendrocytes which assist in the tuning of neural circuitry and the provision of metabolic support to axons. Mouse models and brain tissue samples of long COVID patients have shown extensive white matter-selective microglial and astrocytic reactivity, with subsequent loss of oligodendrocytes and subcortical white matter demyelination (Figure 2). As a result, circuit integrity may be compromised, thereby leading to persisting neurological symptoms [116]. Moreover, novel brain organoid models have demonstrated marked microgliosis 72 h post infection with upregulation of IFN-stimulated genes and microglial phagocytosis leading to engulfment of nerve termini and synapse elimination [117]. This observed postsynaptic destruction may persist along with chronic microglial reactivity to further propagate neurodegeneration in long COVID. Lastly, the most severe long COVID patients exhibited increased levels of tumor necrosis factor receptor superfamily member 11b (TNFRSF11B), an osteoblast-secreted decoy receptor that has been implicated in neuroinflammatory processes and in contributing to microglia overstimulation [90]. Associated with a variety of symptoms, such as cognitive dysfunction, poor psychomotor coordination, and working memory deficits, this mechanism of neural cell reactivity is not specific to COVID-19 and is in fact, strikingly similar to cancer therapy-related cognitive impairment (CRCI) [116]. 

The pathogenesis and neurological manifestations of long COVID implicate disturbances of neuroglial cells with resulting glial cell reactivity that can be localized to specific brain regions, such as the olfactory bulb, brainstem, and basal ganglia [45]. With persistent cytokine abnormalities and brain injury, reactive neuroglia can influence vascular and endothelial function, compromising the integrity of the BBB, and cause neurodegeneration with marked increases in extracellular glutamate leading to toxic hyperexcitability. Reactive microglia respond to increased levels of CCL11 and release microglial cytokines that can damage neural circuitry. This overactive state of microgliosis leads to a decrease in hippocampal neurogenesis, which is linked with deficits in memory and cognitive function, as well as the death of myelinating oligodendrocytes alongside white-matter selective demyelination. In summary, the most consistent neuropathological observation in all autopsy-based studies of COVID-19 patients is the prominent astroglial and microglial over-reactivity. At present, with the exception of a few anecdotal case reports, there are no neuropathological studies of long COVID conditions. Nevertheless, the neuroglial disturbances and ensuing cytotoxicity appear to facilitate persistent inflammation and subsequent axonal dysfunction in the CNS environment, leading to attention deficits, brain fog, fatigue, and anosmia [45,95].

### 3.3. Coaguloapathies and Endotheliopathy-Associated Neurovascular Injury 

COVID-19 is known to increase the risk for hemorrhages, ischemic infarcts, and hypoxic changes in the CNS during the acute phase of infection, implicating endotheliopathy and coagulopathy as important mechanisms of pathogenesis [46]. Although these neurological symptoms are not observed in high frequency among long COVID patients, small vessel thromboses (microclots) and microvascular dysfunction due to persisting mechanisms of endotheliopathy and coagulopathy could account for the neurological symptoms of long COVID that are associated with cerebrovascular disease and hypoxic-neuronal injury (Figure 3) [63]. 

#### 3.3.1. Microclot Formation

A major mechanism of thrombosis in long COVID involves a unique signature of fibrinolysis-resistant, large anomalous amyloid microclot formation present in the serum of patients with long COVID [118]. Thioflavin T staining and microscopy have determined the size of these microclots to reach upwards of 200 µm, which can adequately occlude microcapillaries, reducing cerebral blood flow and causing ischemic neuronal injury [118,119]. Microclot formation occurs due to the binding of the SARS-CoV-2 spike protein with fibrinogen, which causes increased clot density, spike-enhanced release of reactive oxygen species, fibrin-induced inflammation at sites of vascular damage, and delayed fibrinolysis [119,120,121]. Additionally, interaction of the nine-residue segment SK9 located on the SARS-CoV-2 envelope protein with serum amyloid A (SAA) increases fibril formation and stability, thus contributing to the amyloid nature of the microclots [122]. Proteomics pairwise analysis of digested microclot samples from long COVID patients revealed significantly elevated levels of fibrinogen alpha chains and SAA which both contribute to fibrinolysis resistance and subsequent microclot persistence (Figure 3) [118]. The same study also revealed that the inflammatory molecule α2-antiplasmin (α2AP), a potent inhibitor of plasmin, was significantly elevated in microclots from long COVID patients in comparison with patients with acute COVID; likely contributing to an aberrant fibrinolytic system in addition to anomalous microclot formation [118]. 

#### 3.3.2. Antiphospholipid Antibodies

Hypercoagulability can also be precipitated by prothrombotic autoantibody formation in long COVID. Prothrombotic autoantibodies targeting phospholipids and phospholipid-binding proteins (aPLs), including anticardiolipin, anti-beta2 glycoprotein I, and anti-phosphatidylserine/prothrombin, were found to be present in 52% of serum samples of patients hospitalized with acute COVID-19 [123]. It is currently hypothesized that aPLs can form through molecular mimicry, neoepitope formation, or both [124]. The S1 and S2 subunits of the SARS-CoV-2 spike protein could form a phospholipid-like epitope as a mechanism of molecular mimicry or, alternatively, oxidative stress due to SARS-CoV-2 may lead to the conformational change of beta2-glycoprotein I as a way of neoepitope generation. These proposed mechanisms can both result in aPL formation; however, in-vitro experimentation is needed to verify these pathologies in long COVID [124,125]. Antiphospholipid antibodies are then able to cause thrombosis through either the induction of adhesion molecule and tissue factor expression or the upregulation of IL-6, interleukin 8 (IL-8), vascular endothelial growth factor (VEGF), and nitric oxide synthase [124]. However, studies specific to these mechanisms have not yet been undertaken in the setting of long COVID. Alternatively, COVID-19-specific mechanisms of aPL-induced thrombosis include elevated platelet counts and neutrophil hyperactivity [123]. Specifically, IgG was purified from serum of COVID-19 patients with high titers of aPL and added to cultured neutrophils, increasing neutrophil extracellular trap (NET) release (Figure 3) [123]. With the persistence of the S1 subunit of the spike protein within CD16^+^ monocytes for up to 15 months post-infection as an epitope for aPL generation [89], aPL levels can remain elevated in long COVID. 

It is still important, however, to take into account the previously mentioned insignificant autoantibody generation to the exoproteome in long COVID patients when considering the role of aPL in disease pathogenesis [64]. Thus, the interplay between anomalous microclot formation, fibrinolytic system dysfunction, and possible aPL formation likely contribute to persistent coagulopathies leading to ischemic neuronal injury in long COVID.

#### 3.3.3. Endotheliopathy

Persistent endotheliopathy, independent of the acute COVID-19 response, has been implicated in BBB disruption and neurovascular injury. Levels of plasma markers for endotheliopathy, including von Willebrand factor (VWF) antigen, VWF propeptide, soluble thrombomodulin, and endothelial colony-forming cells, remained elevated in a cohort of patients assessed at a median of 68 days post-infection [126,127]. This prolonged endotheliopathy in long COVID can be attributed to the sustained effect by tumor necrosis factor α (TNF-α) and interleukin 1β (IL-1β) proinflammatory cytokines [128], complement activation by immunoglobulin complexes [129], oxidative stress evidenced by elevated malondialdehyde levels [130], or by direct viral invasion of endothelial cells [113]. With regards to the immune-mediated processes, although cytokines themselves can directly activate endothelial cells, immune complexes positive for IgG and immunoglobulin M (IgM) at the vascular wall of post-mortem tissue of patients with acute COVID-19 were co-localized with membrane attack complexes (MAC) composed of activated C5b-9 complement factors [129]. The presence of MAC, paired with the previously mentioned evidence of autoantibodies to ACE2 receptors on endothelial cell surfaces [99], could very well lead to endothelial cell death. Additionally, in regards to viral invasion of endothelial cells, in vitro and in vivo experiments have elucidated the ability of SARS-CoV-2 protease Mpro to cleave NF-κB essential modulator (NEMO) in endothelial cells (Figure 3), resulting in cell death, empty basement membrane tube formation (also known as string vessels), and BBB dysfunction in mice [130]. 

#### 3.3.4. Blood–Brain Barrier Disruption

BBB dysfunction has been hypothesized to be at the center of the mechanisms of long-term COVID complications. BBB alterations in permeability after addition of extracted SARS-CoV-2 spike protein have been observed in microfluidic models, likely due to endotheliopathy from a pro-inflammatory response [131]. The resulting BBB permeability and microvascular injury have been indicated by perivascular leakage of fibrinogen and persistent capillary rarefaction in an autopsy and a sublingual video microscopy study of long COVID patients, respectively [129,132]. BBB dysfunction can then allow for infiltration of immune cells and cytokines from the systemic circulation that can then propagate neuroinflammation mechanisms in the CNS. Notably, the same autopsy study revealed perivascular invasion of CD68^+^ macrophages and CD8^+^ T cells (Figure 3) along with notable reactive astrogliosis which might serve a currently unidentified role in perpetuating BBB leakage [129]. This loss of BBB function is thought to be more pronounced in areas of the cerebellum and brainstem, where most pathological abnormalities have been found with prominent hypometabolism in the bilateral pons, medulla, and cerebellum of long COVID patients (Figure 1) [45,46,133]. The increased permeability at the blood and CNS interface could allow for microglial activation by systemic inflammation evidenced by the presence of microglial nodules associated with neuronophagia and neuronal loss in the hindbrain of patients with acute COVID-19 [129], likely accounting for the persisting hypometabolic pathology. Subsequent neuronal degeneration and brainstem dysfunction could explain the similarities between long COVID symptoms and ME/CFS since the association between severity of ME/CFS symptoms and brainstem dysfunction has been elucidated in previous imaging studies [134,135,136,137]. Together, all endothelial-associated mechanisms can lead to the spread of inflammatory cytokines and immune cells into the CNS, infected leukocyte extravasation across the BBB, and microhemorrhage, ultimately contributing to underlying neurological and cognitive symptoms in long COVID. Current interventions that aim to ameliorate risk of thrombotic complications are not COVID-19 specific, however, due to the implication of SARS-CoV-2 Mpro in endotheliopathy, the role of nirmatrelvir or Paxlovid as Mpro inhibitors could possibly lessen BBB dysfunction. Still, a pharmacological challenge remains in demonstrating the benefit of traditional anticoagulation in patients with long COVID. 

## 4. Conclusions

Neurological manifestations of long COVID exist as a major complication of COVID-19 post-infection, affecting up to one third of patients with COVID symptoms lasting longer than four weeks. Although SARS-CoV-2 neurotropism, viral-induced coagulopathy, endothelial disruption, systemic inflammation, cytokine overactivation and neuroglial dysfunction have been hypothesized as mechanisms associated with pathogenesis of long COVID condition, further clinical, neuropathological, and experimental models are needed to address many of the unknown questions about pathogenesis. Similarly, current and potential therapeutics to target these hypothesized pathogenic mechanisms using anti-inflammatory, anti-viral, and neuro-regenerative agents are potentially able to reverse neurological sequelae but still require well designed clinical trials studies to prove their efficacy. 

## Figures and Tables

**Figure 1 cells-12-00816-f001:**
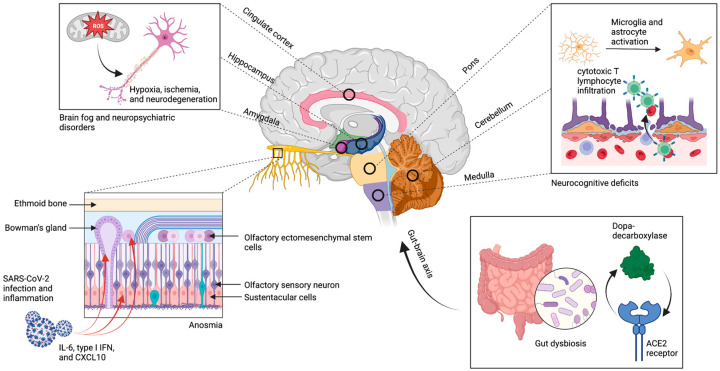
Neuroinvasion and persistent viral shedding. SARS-CoV-2 employs the ACE2 receptor to invade the stem cells, perivascular cells, sustentacular cells, and Bowman’s gland cells in the olfactory epithelium; this leads to chronic thinning of filia and loss of olfactory bulb volume. Additionally, there is an association between areas of hypometabolism in the cortex, cerebellum, and brainstem with the spatial distribution of ACE2 receptors, though there is little evidence for direct neuroinvasion in these areas. Rather, it is hypothesized that these regions experience elevated levels of microglial activation, cytotoxic T lymphocyte infiltration, oxidative stress, and neurodegeneration and demyelination secondary to neuroinvasion. These mechanisms likely persist due to the chronic presence of viral shedding specifically in the gastrointestinal tract where there exists ACE2 co-regulation of DDC and involvement of the dopamine metabolic pathway. Figure was created with the BioRender software.

**Figure 2 cells-12-00816-f002:**
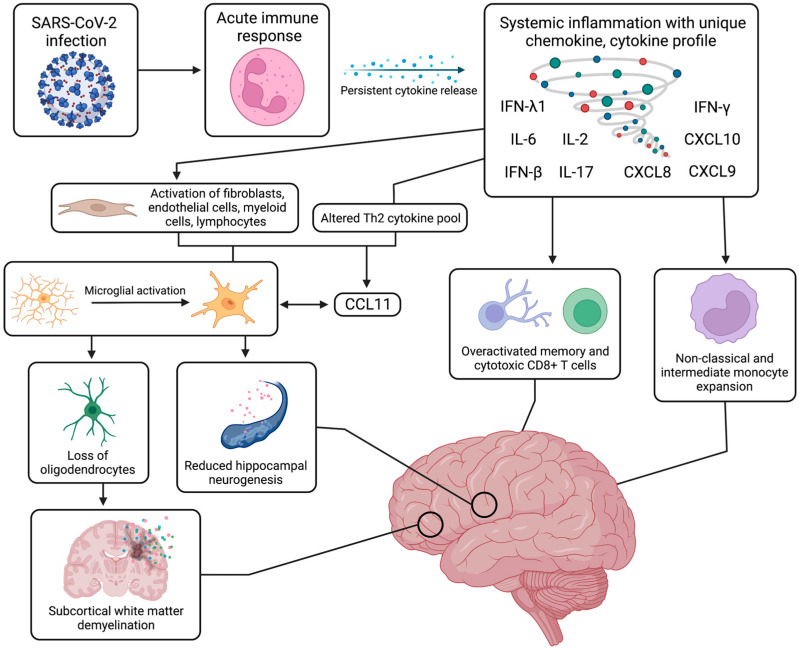
Systemic and neurological immune response. The systemic immune and inflammatory response to SARS-CoV-2 infection can continue for months after the acute recovery phase, inducing a state of persistent systemic inflammation with upregulated cytokines, such as IFN-β, IFN-λ1, IFN-γ, IL-2, IL-6, IL-17, CXCL8, CXCL9, and CXCL10. This prolonged cytokine release has been linked to activation of specific immune cell populations, such as non-classical and intermediate monocytes, as well as other cell types, such as fibroblasts and myeloid cells. From an aberrant Th2 cytokine pool, production of CCL11 is induced and leads to neuroinflammation with activation of resting microglia, which can further release increased levels of CCL11. This microglial reactivity can in turn cause reduced hippocampal neurogenesis, loss of myelinating oligodendrocytes and oligodendrocyte precursors, and ensuing subcortical white matter demyelination. These systemic and neurological mechanisms have been strongly associated with a range of cognitive impairments and neuropsychiatric symptoms. Figure was created with the BioRender software.

**Figure 3 cells-12-00816-f003:**
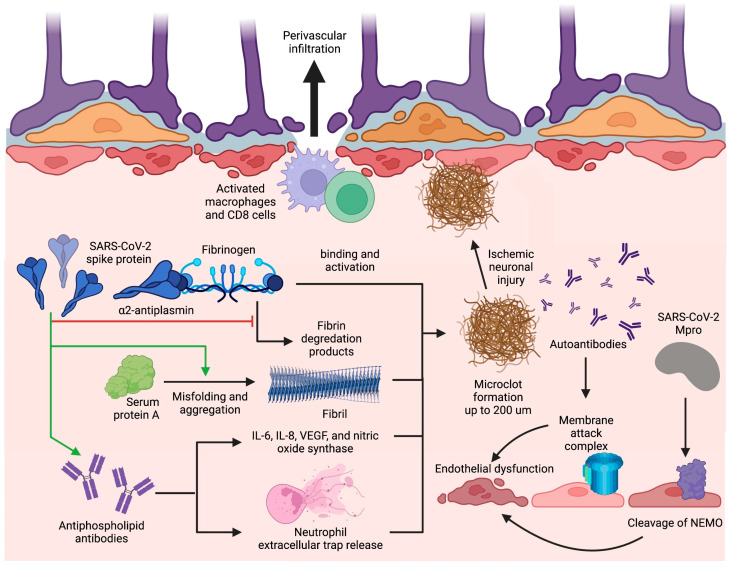
Blood–brain barrier disruption and microclot formation. SARS-CoV-2 can cause increased microclot formation through spike protein interactions with fibrinogen and serum protein A that promote fibril formation and resist fibrinolysis. Antiphospholipid antibodies are also present in long COVID and can precipitate microclot formation through IL-6, IL-8, VEGF, nitric oxide synthase, and NET release. These microclots also contain α2AP which inhibit plasmin and thus prevent the degradation of fibrin, further contributing to their fibrinolysis-resistant nature. Additionally, SARS-CoV-2 can induce BBB disruption through Mpro cleavage of NEMO in endothelial cells leading to cell death and string vessel formation. Figure was created with the BioRender software.

**Table 1 cells-12-00816-t001:** Published trials on interventions for neurological symptoms of long COVID.

		Population Characteristics						
Reference/Country	Study Design	Sample Size (n)	Inclusion Criteria/Long COVID Diagnosis	Acute COVID Diagnosis	Comorbidities	Intervention	Comparator	Outcomes
Xie et al. [69] United States	Quasi-experimental with no blinding	56,340	At least one post-acute sequela from a set of 12 prespecified post-acute sequelae of SARS-CoV-2 infection 30 days after positive test	Laboratory confirmed	15.02% had cancer, 22.95% had chronic lung disease, 8.09% had dementia, 33.96% had type 2 diabetes mellitus, 30.92% had cardiovascular disease, 80.99% had hyperlipidemia	Nirmatrelvir	None	Reduced risk of fatigue and neurocognitive impairment
Singh et al. [70] India	Unclear	120	Mild to moderate COVID-19	Positive RT-PCR test	Not mentioned	Nasal fluticasone spray	None	Significant improvement in recognizing odors
Abdelalim et al. [71] Egypt	Randomized double blind	100	Unclear	Positive rRT–PCR test by nasopharyngeal swab	16% had diabetes mellitus, 14% had hypertension	Mometasone furoate nasal spray and olfactory training	Only olfactory training	Treatment showed no significant improvement in recovery rates and duration of anosmia
O’Kelly et al. [72] Ireland	Quasi-experimental with no blinding	52	Ongoing symptoms 12 or more weeks after initial infection	Laboratory Confirmed	9.6% had hypertension, 9.6% had dyslipidemia, 3.8% had diabetes mellitus	Low dose naltrexone	None	Improvement in activities of daily living, energy levels, pain levels, levels of concentration and sleep disturbance
Mohamad et al. [73] Egypt	Randomized single blind parallel design	40	Post-COVID olfactory loss verified by endoscopic examination with anosmia lasting for at least two weeks	Unclear	Not mentioned	Insulin films	Plain films	Significant improvement in olfactory detection scores and olfactory values discrimination
Rezaeian [74] Iran	Randomized double blind	38	Hyposmic patients with a Connecticut Chemosensory Clinical Research Center (CCCRC) score between 2 and 5.75	NA	7.9% had hypertension, 2.6% had cardiovascular disease	Insulin therapy	Normal saline therapy	Administration of intranasal insulin significantly improved hyposmia with higher mean CCCRC scores
Glynne et al. [75] United Kingdom	Quasi-experimental with no blinding	49	Either PCR, serological evidence, or suffered acute illness	Positive RT–PCR test	Not mentioned	Loratadine 10 mg two times per day or fexofenadine 180 mg two times per day and loratadine 10 mg two times per day or fexofenadine 180 mg two times per day	Supportive care	Decrease in fatigue, neurologic, and neuropsychiatric symptoms among many others

**Table 2 cells-12-00816-t002:** Ongoing trials on interventions for neurological symptoms of long COVID.

		Population Characteristics						
Trial Registration Number/Country	Study Design	Recruitment Target (n), Age Range (Years)	Definition of Long COVID	Method of Acute COVID-19 Diagnosis	Comorbidities	Intervention	Comparator	Neurological Outcomes
NCT0557666 United States	Randomized double blind	200, >18	Post-COVID-19 symptoms persisting greater than three months with at least two post-COVID symptoms of moderate or severe intensity	Self-reported history of confirmed COVID-19 infection preceding post-COVID symptoms	Excluded individuals with severe liver disease, HIV infection, suspected or confirmed pregnancy or breastfeeding, or other medical conditions that would compromise patient’s safety or compliance	Nirmatrelvir with Ritonavir (Paxlovid)	Placebo with Ritonavir	Severity of brain fog, PROMIS cognitive function abilities score
NCT05430152 Canada	Randomized double blind	160, 19–69	Symptoms lasting three to six months post-acute infection	Positive test result or clinical confirmation by a physician	Excluded participants with any use of opioid medications	Low dose naltrexone	Placebo	Changes in fatigue intensity score, pain severity score, and symptom severity score
NCT05597722 United States	Randomized open label	120, 21–65	COVID-symptoms that persist three months or longer and a moderate cognitive impairment (MOCA) score ≤18 for at least three months	Positive PCR or home antigen test	Excluded individuals with pre-existing cardiac or kidney condition, severe hypertension, glaucoma, hyperthyroidism, and advanced arteriosclerosis	Dextroamphetamine-Amphetamine	Stimulant medication	Changes in cognitive impairment
NCT0494412 United States	Randomized quadruple blind	70, 18–75	Confirmed SARS-CoV-2 infection by PCR at least 24 weeks prior to baseline with a raw score of at least a score of 21 on PROMIS Fatigue SF 7a at screening	Laboratory-confirmed SARS-CoV-2 PCR test	Excluded individuals with orthostatic hypertension, tachycardia, use of sedating medications, lab abnormalities that may cause fatigue, history of anaphylaxis, previous chronic fatigue syndrome, previous fibromyalgia, previous lupus, previous Sjogren’s syndrome, or diagnosis of sleep apnea	RSLV-132	Placebo	PROMIS fatigue 7a T-score, FACIT fatigue questionnaire, severity of brain fog, performance on concentration task
NCT05350774 United States	Randomized double blind	60, >18	Persistent neurological symptoms exceeding 12 weeks from acute infection	Patient reported positive antigen test for SARS-CoV-2 followed by confirmatory nucleocapsid antibody testing or a positive SARS-CoV-2 PCR test result	Excluded individuals with ventricular arrhythmias, coronary artery disease, or a condition prior to COVID-19 infection that would confound interpretation	IV immunoglobulin, IV methylprednisone	Placebo	Proportion of participants with clinically meaningful improvement in the Health Utilities Index Mark 3 (HUI3) four weeks after start of study
NCT0561858 United States	Randomized quadruple blind	50, 18–80	Reported fatigue and/or brain fog for less than four weeks prior to enrollment on questionnaire after SARS-CoV-2 infection	Documented or self-reported positive test for COVID-19 less than four weeks prior to enrollment	Excluded individuals with current alcohol abuse, history of fibromyalgia, chronic fatigue syndrome, or progressive cognitive disorder, any active medical, psychiatric, or social problems that would interfere with study procedure, or use of any tobacco, marijuana, or illicit drug products	Lithium	Placebo	Severity of brain fog, frequency of anxiety, frequency of headaches, severity of insomnia, change in sense of smell and taste, and performance on cognitive tests
NCT05104424 Saudi Arabia	Randomized open label	44, >18	Unclear	Unclear	Excluded patients with any nasal deviation, congenital anomalies, or allergic sinusitis	Intranasal insulin, zinc, gabapentin, ice cube stimulation	Zinc only	Improvements in hyposmia (Sniffin Sticks test), dysgeusia, and self-assessment of hyposmia (Dynachron-olfaction questionnaire)
NCT04997395 United Kingdom	Open Label	12, ≥18	Confirmed by GP triage clinic and assessment by a long COVID clinic. Participant undertook the clinical assessment and investigations as recommended by the NICE guidance on long COVID	Unclear	Excluded individuals with ongoing mental and/or psychiatric illnesses/disorders that require active treatment during the trial period, use of anti-coagulant drugs, use of cannabinoids or cannabinoid-based medicine within three months prior to study, known dependence on cannabis, alcohol, or any other drug, history of chronic liver failure, history of allergy to tree nuts	MediCabilis Cannabis sativa 50	Discontinuing cannabidiol for three-weeks following 21 week period of taking cannabidiol	Changes in cognition, fatigue, self-reported quality of life, pain score, and anxiety/depression

## Data Availability

No new data were created or analyzed in this study. Data sharing is not applicable to this article.

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
