# Peer review of "Pathogenesis Underlying Neurological Manifestations of Long COVID Syndrome and Potential Therapeutics"

_cells, 2023, doi:10.3390/cells12050816_

Round 1

Reviewer 1 Report

In this review paper titled 'Pathogenesis Underlying Neurological Manifestations of Long  COVID Syndrome and Potential Therapeutics', the authors Leng et al discuss the current understanding of the pathology of long-COVID potential therapeutics, with a focus on the neuropathology. The authors did an excellent job offering a comprehensive overview of the literature. There are a couple of prominent review papers the author might want to include: 

1. Peluso, M. J.; Deeks, S. G., Early clues regarding the pathogenesis of long-COVID. Trends Immunol. 2022, 43 (4), 268-270.

2. Monje, M.; Iwasaki, A., The neurobiology of long COVID. Neuron 2022, 110 (21), 3484-3496.

Author Response

Thank you for bringing our attention to these important review papers. Peluso et al.’s review presents a similar discussion about long-COVID’s risk factors, pathogenesis, and underlying mechanisms, such as an impaired antiviral response and persistent inflammatory state. We have included citations to their paper in the “Risk Factors” section because they also mention the risk of diabetes mellitus and Ebstein Barr virus reactivation in developing long-COVID. Also, we have referenced Peluso et al. in the “Autoantibody Generation” section as the authors describe how residual inflammation may lead to the generation of various downstream autoimmune processes.

Monje et al.’s review validates many of the mechanisms we found to be important as well. We have added citations to their paper in the “Reactivation of Herpesviruses” section since they also call attention to the parallel between the immune response to the reactivation of EBV and ME/CFS like symptoms. We also included Monje et al. as a citation in the introduction of the “Coaguloapathies and Endotheliopathy-Associated Neurovascular Injury” section since they also emphasize the importance of coagulopathy and draw similar conclusions on its effect on cerebral vasculature. 

Reviewer 2 Report

Pathogenesis Underlying Neurological Manifestations of Long-COVID Syndrome and Potential Therapeutics

Leng et al.

The present manuscript focus to review different mechanisms inducing long-COVID and therapeutic strategies used to reduce this pathological condition. This is an interesting review about a neuropathological severe condition that results as consequence of the SARS-CoV-2 infection. The manuscript is well organized. However, it is necessary to consider major concerns.

- Studies indicate that the susceptibility to SARS-CoV-2 infection and the outcome of the COVID-19 disease can depend on ACE2 expression and/or polymorphism. Therefore, it is essential to describe the possible participation of polymorphisms in the development and expression of long-COVID.  

-Mast cells are important immune modulators inducing protective effects and pathogenic effects as consequence on viral entry of SARS-CoV-2. The authors have to consider that mast cells are in contact with the blood-brain barrier and their disfunction can affect the brain. Indeed, there are studies focused to discuss the role of mast cells in long-COVID (see Arun S, Storan A, Myers B. Mast cell activation syndrome and the link with long COVID. Br J Hosp Med (Lond). 2022 Jul 2;83(7):1-10. doi: 10.12968/hmed.2022.0123. Epub 2022 Jul 26. PMID: 35938771). According to this issue, the role of mast cell in long-COVID and the utility of mast cell stabilizers as therapy in long-COVID has to be included in the manuscript.

-In the section 3 “Mechanisms of Neurological Long-COVID and Review of Therapeutics” I suggest including the use of cannabidol as therapy for long-COVID. Several studies support that cannabidiol downregulates proteins responsible for viral entry and inhibits SARS-CoV-2 replication. In addition, cannabidiol induces neuroprotective effects. Obviously, further clinical studies are necessary to prove the suitability of cannabidiol for the treatment of long-COVID.

-Describe abbreviations such as ICU

Author Response

Response to Reviewer 2 Comments

Point 1: Studies indicate that the susceptibility to SARS-CoV-2 infection and the outcome of the COVID-19 disease can depend on ACE2 expression and/or polymorphism. Therefore, it is essential to describe the possible participation of polymorphisms in the development and expression of long-COVID. 

Response 1: We have included in the first paragraph of the section “Viral Neuroinvasion and Persistent Viral Shedding” a discussion on the current literature available relating ACE2 and TMPRSS2 polymorphisms and expression to the development of long-COVID symptoms. The current literature seems limited and no conclusions can be drawn. 

Point 2: Mast cells are important immune modulators inducing protective effects and pathogenic effects as consequence on viral entry of SARS-CoV-2. The authors have to consider that mast cells are in contact with the blood-brain barrier and their disfunction can affect the brain. Indeed, there are studies focused to discuss the role of mast cells in long-COVID (see Arun S, Storan A, Myers B. Mast cell activation syndrome and the link with long COVID. Br J Hosp Med (Lond). 2022 Jul 2;83(7):1-10. doi: 10.12968/hmed.2022.0123. Epub 2022 Jul 26. PMID: 35938771). According to this issue, the role of mast cell in long-COVID and the utility of mast cell stabilizers as therapy in long-COVID has to be included in the manuscript.

Response 2: We appreciate your comments on including a brief discussion of the role of mast cells in long-COVID and mast cell stabilizers in treating long-COVID. We have included the above citation along with several other papers that discuss mast cell activation in the long-COVID setting. Weinstock et al. presents a similar discussion as Arun et al. in drawing similarities between long-COVID and mast cell activation syndrome (MCAS), namely in how mast cell overactivation may contribute to ongoing systemic inflammation and CNS disturbances. Glynne et al. note that antihistamine treatment led to an improvement in symptoms in long-COVID patients, warranting further exploration of mast cell stabilizers as potential therapeutics.

Point 3: In the section 3 “Mechanisms of Neurological Long-COVID and Review of Therapeutics” I suggest including the use of cannabidol as therapy for long-COVID. Several studies support that cannabidiol downregulates proteins responsible for viral entry and inhibits SARS-CoV-2 replication. In addition, cannabidiol induces neuroprotective effects. Obviously, further clinical studies are necessary to prove the suitability of cannabidiol for the treatment of long-COVID.

Response 3: Thank you for drawing our attention to the use of cannabidiol as a therapy for long-COVID. At the end of the “Related Long-COVID Therapeutics: Antivirals” subsection within the “Mechanisms of Neurological Long-COVID and Review of Therapeutics” section, we have added several studies. These studies demonstrate how cannabidiol can inhibit SARS-CoV-2 replication and viral invasion, as well as induce neuroprotective effects. Thank you again.

Point 4: Describe abbreviations such as ICU

Response 4: The full term (intensive care unit) for the abbreviation ICU has been included at the first instance of its mention in the manuscript. Thank you for pointing this out.

Round 2

Reviewer 2 Report

The authors addressed all the comments. No further comments.